# Are There Any Beneficial Effects of *Spirulina* Supplementation for Metabolic Syndrome Components in Postmenopausal Women?

**DOI:** 10.3390/md18120651

**Published:** 2020-12-17

**Authors:** Elena Bobescu, Andreea Bălan, Marius Alexandru Moga, Andreea Teodorescu, Maria Mitrică, Lorena Dima

**Affiliations:** 1Department of Medical and Surgical Specialties, Faculty of Medicine, Transilvania University of Brasov, 500019 Brasov, Romania; elena.bobescu@unitbv.ro (E.B.); mogas@unitbv.ro (M.A.M.); maria.mitrica@unitbv.ro (M.M.); 2Department of Fundamental, Prophylactic and Clinical Sciences, Faculty of Medicine, University Transilvania Brasov, 500019 Brasov, Romania; andreea.teodorescu@unitbv.ro (A.T.); lorena.dima@unitbv.ro (L.D.)

**Keywords:** *Spirulina*, menopause, metabolic syndrome, dyslipidemia, insulin resistance, obesity, blood pressure

## Abstract

*Spirulina* is a phytosynthetic filamentous cyanobacterium with microscopic dimensions, which naturally grows in the highly-salted alkaline lakes of Africa, Mexico, America, and Asia. Several bioactive peptides extracted from *Spirulina* were demonstrated to possess antimicrobial, antiviral, antitumor, immunomodulatory, antiallergic and antihypertensive properties. It has been reported that the consumption of *Spirulina* could prevent or manage metabolic syndrome components. In women, metabolic disorders are more prevalent during menopause. Postmenopausal women present higher waist circumference, increased blood pressure, hypertriglyceridemia, hyperglycemia, and decreased HDL-cholesterol values, leading to an increased risk of cardiovascular events. Therefore, in order to prevent cardiovascular diseases, it is essential to manage the components of the metabolic syndrome during the postmenopausal period. As recent reports indicated the efficiency of *Spirulina* supplementation in the management of the metabolic syndrome components, our study aims to review all the clinical trials conducted on this topic. Our main objective is to have a better understanding of whether and how this cyanobacterium could manage the abnormalities included in the metabolic syndrome and if it could be used as a therapeutic approach in postmenopausal women with this condition. We selected relevant articles from PubMed, Google Scholar and CrossRef databases, and a total number of 20 studies met our criteria. All included clinical trials indicated that *Spirulina* has positive effects in managing metabolic syndrome components. *Spirulina* is a valuable cyanobacterium that can be used as a food supplement for the management of metabolic syndrome, and it is able to reduce the risk of cardiovascular events. The optimal dose and period of administration remain a debated subject, and future investigations are required. Considering the beneficial effects reported against each component of the metabolic syndrome, *Spirulina* could also be effective in the postmenopausal period, when this syndrome is the most prevalent, but there is a strong need for human clinical trials in order to sustain this observation.

## 1. Introduction

*Spirulina,* also known as *Arthrospira platensis*, is a phytosynthetic filamentous cyanobacterium with microscopic dimensions, which showed intense biomass productivity, with the highest CO2 fixation rate [1]. *Spirulina* naturally grows in the high-salted alkaline lakes of Africa, Mexico, America and Asia, but at present, it is commercially produced all over the world [2]. There is a large number of *Spirulina* species, but only three of them were intensively investigated: *Spirulina platensis, Spirulina fusiformis* and *Spirulina maxima* [3].

*Spirulina* has been suggested as an eco-friendly cyanobacterium, which is widely consumed as a nutritional supplement for its multiple beneficial effects on humans and animals’ health. This cyanobacterium usually has rapid growth, does not need fertile land for its development [4], and due to its high nutritional value and protein content, it has been proposed for the improvement of meat quality [5]. During the Aztec civilization, more than 400 years ago, *Spirulina* was consumed as food by the Mayas and Kanembu [3].

*Spirulina* is a blue-green cyanobacterium that has been used for ancient times due to its extraordinary nutritional profile. It abounds in proteins, which represent almost 60–70% by dry weight, and also contains all the essential amino acids, high amounts of carotenoids (6.25%) [6], essential fatty acids (linoleic, gamma-linolenic and palmitic acid), vitamin E, C and selenium [7]. Due to its concentrated nutrition, *Spirulina* has become in the last years one of such nutraceutical food for managing various health issues [3].

Several bioactive peptides extracted from this cyanobacteria were demonstrated to possess anti- antimicrobial, antiviral, antitumor, immunomodulatory, antiallergic and antihypertensive properties (Figure 1) [8]. Furthermore, phycobiliprotein C-phycocyanin and other phenolic phytochemicals from *Spirulina* exert strong antioxidant and anti-inflammatory effects [9]. According to previous reports, alternative drugs derived from natural products are safer as anti-inflammatory agents than pharmaceutical drugs because they are generating fewer side effects [10].

*Spirulina* is being used as a therapeutic tool in the management of metabolic syndrome, which is defined as multiple interconnected metabolic abnormalities, including dyslipidemia, glucose intolerance, obesity, hypertension, and prothrombotic state [11]. The worldwide prevalence of this metabolic disorder is about 84%, and it is in a continuous ascent, depending on the region, environment, and population characteristics [12].

Menopause is a physiological period in a woman’s life and represents the ovarian function’s permanent cessation [13,14]. During this period, the prevalence of metabolic syndrome increases. Postmenopausal women present higher waist circumference, increased blood pressure, hypertriglyceridemia, hyperglycemia, and decreased HDL-cholesterol values, leading to an increased risk of cardiovascular events [15,16]. Therefore, in order to prevent cardiovascular diseases, it is essential to manage the components of the metabolic syndrome during the postmenopausal period [17,18]. As recent reports indicated the efficiency of *Spirulina* supplementation in the management of metabolic syndrome components, our study aims to review all the clinical trials conducted on this topic. Moreover, we aim to create a common denominator of them in order to have a better understanding of whether and how this cyanobacterium could improve metabolic abnormalities. Regarding the effects of *Spirulina* in postmenopausal women with metabolic syndrome, in the literature, there is a lack of clinical trials focused only on this category of subjects. Through our study, we want to gather all the evidence of the effectiveness of *Spirulina* against the components of the metabolic syndrome in different categories of patients and to pave the way for future studies that would address the effects of this cyanobacterium strictly in postmenopausal women.

## 2. Metabolic Syndrome and Menopause

The incidence of metabolic syndrome (obesity, hyperglycemia, dyslipidemia, and hypertension) substantially increases during menopause, in comparison with the menopausal transition. A study including Iranian women has shown that the prevalence of metabolic syndrome was 53.5% in postmenopausal subjects and 18,3% during the menopausal transition [20].

Obesity and obesity-related disorders represent a significant public health concern. The real reason for increasing obesity during menopause is not clear for the moment, but many genetic and environmental factors were involved in the pathophysiology of adipose cell accumulation [21]. Some studies reported that sudden estrogen withdrawal might be the most crucial obesity-triggering factor [22]. According to these findings, estrogen deficiency enhances metabolic abnormalities, leading to diabetes mellitus type 2, metabolic syndrome and other cardiovascular events [23].

In the USA, the prevalence of visceral obesity is almost double in comparison with general obesity. Perimenopause is mainly associated with fat redistribution, resulting in the transition from the gynoid to the android fat distribution pattern [24]. The visceral fat is able to secrete adipokines, which are small molecules closely associated with metabolic syndrome. Adipocytes regulate the production of various cytokines, which control the hunger center, the satiety center and modulate the energy repartition in different tissues [25].

Human visceral and subcutaneous adipose tissues express estrogen receptors (ER), such as ERα and ERβ [26]. In brown adipose tissue, ERβ has not been identified [27]. ERα influences the activity of the adipocytes and the distribution of adipose tissue. Women that lack ERα develops diabetes mellitus type 2 and increased insulin resistance [28]. In conclusion, estrogen is essential for the regulation of adipocytes metabolism and for the pattern of adipose depots. In these conditions, the sudden estrogen decrease induces visceral obesity, metabolic abnormalities and increases the risk for cardiovascular events.

Although the association between central obesity and the alterations of the lipid metabolism in elderly women is well known, the pathophysiological mechanism is not well described yet. Increased visceral depots of adipose tissue usually associate increased free fatty acid levels, insulin resistance, and decreased levels of adiponectin. The consequence consists of increased secretion of apolipoprotein B (apoB) and increased hepatic lipase activity, leading to hypertriglyceridemia [23].

During menopause, the levels of total cholesterol, triglycerides, and LDL-cholesterol significantly increase, while HDL-cholesterol decreases. The composition of the LDL-cholesterol particles also changes in the postmenopausal period. The preponderance of small and dense LDL-cholesterol molecules, with increased atherogenic potential, is associated with an increased risk of cardiovascular diseases [29,30]. Regarding the levels of triglycerides, Poehlman et al. [31] reported that the transition to the postmenopausal period is associated with a 16% increase in triglyceride levels. Moreover, studies have shown that triglyceride level exponentially increases with the transition to menopause, the highest values being reached in the postmenopausal period [32]. Triglycerides are positively correlated with abdominal fat and insulin resistance [32].

Hypertension is, by far, the most important risk factor for coronary heart diseases (CHD), especially after the age of 63 years [33]. Rising values of the blood pressure during menopause are mainly induced by the increase of vascular stiffness of the great arteries, combined with atherosclerosis, secondary to the lipid metabolism abnormalities [34]. Moreover, during this period, the decline in the estrogen/androgen ratio triggers the production of vasoconstrictive factors, such as endothelin. It significantly decreases the vasorelaxant effects of estrogen on great vessels [35]. In addition, the activation of the renin–angiotensin–aldosterone system (RAA) and the increase of oxidative stress play important roles in the development of hypertension during menopause [35].

Considering that is no animal model of naturally occurring postmenopausal high blood pressure discovered until the present, the elucidation of all the mechanisms involved in hypertension development in menopause dwindled down [36]. In the postmenopausal period, the oxidative stress increases, leading to reduced levels of plasma glutathione peroxidase and increased levels of superoxide dismutase, which triggers the increase of blood pressure. Moreover, in older women, endothelin, a molecule that stimulates oxidative stress by upregulating the subunits of NAD(P)H oxidase, is significantly higher in comparison with younger women.

Many women gain weight or become obese after menopause. This weight gain, associated with the dramatic loss of estrogen levels, usually leads to hypertension and increased incidence of type 2 diabetes mellitus. Previous results have revealed the protective role of estrogen against type 2 diabetes mellitus and metabolic syndrome [37]. Estrogen is able to regulate insulin action by regulating oxidative stress, which contributes to insulin resistance, or by acting directly on insulin-sensitive tissues. It has been reported that ERα exerts positive effects on GLUT4 expression and global insulin action [38].

Obesity and increased visceral adiposity are the main risk factors for type 2 diabetes mellitus in this period [39]. It has been reported that insulin resistance observed in type 2 diabetes mellitus usually occurs in women genetically predisposed when compounded with increased visceral adiposity [40].

All these metabolic changes characteristic for the postmenopausal period should be timely diagnosed and treated to prevent complications. Furthermore, a moderate calorie intake associated with several lifestyle changes (physical exercises, administration of dietary supplements, etc.) could prevent proatherogenic changes and obesity during menopause [41], leading to a great quality of life during this period.

Figure 2 illustrates the mechanisms of the metabolic abnormalities in postmenopausal women.

## 3. Chemistry and Biochemistry of Spirulina

The composition of *Spirulina* mainly consists of carbohydrates, proteins, and lipids [42]. Usually, proteins are associated with biosynthesis and cellular division, while carbohydrates and lipids mainly serve as intracellular reservoirs of energy [43]. Although there are thousands of *Spirulina* strains available in cultures, significant differences were observed in their main macromolecular composition across the different phyla of this cyanobacterium [44]. A pioneering study conducted in 1961 [45] reported that *Spirulina* contains 39% proteins, 23% carbohydrates and 8% lipids, on average. More recently, a meta-analysis including 130 studies has shown that the median macromolecular composition of *Spirulina* consists of 32.2% proteins, 15% carbohydrates, 17.3% lipids and 17.3% ash [46]. It has been reported that in comparison with eukaryotic microalgal phyla, cyanobacteria possess lower lipid and ash, and higher protein and carbohydrate, as percent dry weight [46]. The composition of *Spirulina* can be strongly affected by both environmental conditions and phyla [42].

The most well-described alteration induced by inappropriate cultivation conditions consists of nitrogen starvation. This event is leading to lipids accumulation, but the composition of the fatty acids remains unaffected. Nitrogen starvation suspends the synthesis of long-chain fatty acids in *Spirulina* platensis, and when the quantity of nitrogen is limited, this cyanobacterium stores carbon [47]. Additionally, the phosphorous limitation is leading to increased carbohydrate and lipid content [48].

Regarding the lipid content of Spirulina, it consists of linoleic acid, γ-linolenic acid, and fatty acids. Lower concentrations of polyunsaturated fatty acids, docosahexaenoic acid, and eicosapentaenoic acid have also been reported [49].

*Spirulina* has the highest protein and essential amino acid content ever found in one cyanobacterium [50]. *Spirulina* abounds in phycobiliproteins such as phycoerythrin, phycocyanin and allophycocyanin. Phycoerythrin is a fuchsia pigment, while phycocyanin and allophycoerythrin are bright blue pigments. These pigments are usually used in the food industry, pharmaceutical industry, or cosmetic industry, depending on their purity [42]. Another pigment extracted from *Spirulina* is phycocyanobilin [51]. It is part of the phycobilins, has a similar structure with biliverdin, and because of these structural similarities, it can be metabolized by biliverdin reductase. Phycocyanobilin is able to exert potent antioxidant and anti-inflammatory properties [52].

*Spirulina* also contains other constituents, such as vitamins (B vitamin complex) and minerals: selenium, iron, potassium, calcium, zinc, magnesium, etc. [53]. Most cyanobacteria contain pseudo-vitamin B12, an inactive corrinoid, but *Spirulina* also contains active vitamin B12 [54].

Other compounds described in the composition of *Spirulina* are carotenoids, a class of pigments that act as provitamin A. From these, zeaxanthin, β-carotene, and astaxanthin were recently described [49]. Chlorophyll is a green pigment in Spirulina, with antioxidant, anti-inflammatory, and antibacterial properties [55,56].

The chemical structure of *Spirulina* consists of multicellular cylindrical trichomes. This cyanobacterium is able to create a unique helical shape. Different species of *Spirulina* can be identified through the length and size of the helix [57]. Overall, these cyanobacteria contain a wide range of essential components for a healthy and balanced diet, and dietary supplementation with *Spirulina* could have many health benefits by mitigating a wide number of pathologies.

## 4. Material and Methods

This article is a literature review based on *Spirulina* supplementation effects against the components of the metabolic syndrome. Given the high prevalence of this syndrome in the postmenopausal period, we aimed to investigate whether and how the dietary supplementation with this cyanobacterium could improve the metabolic abnormalities and if it could also be effective in menopausal women. We selected the relevant studies from PubMed, Google Scholar and CrossRef databases, using the following Medical Subject Headings (MeSH) keywords: “Spirulina”, “menopause”, “metabolic syndrome”, “dyslipidemia”, “insulin resistance”, “obesity”, “blood pressure”. Two authors separately identified the relevant papers and selected them based on the following inclusion criteria: full-text original articles written in English. Only human clinical trials were considered for this review. The exclusion criteria consisted of papers written in languages other than English, abstracts, duplicate papers, and preclinical studies conducted on animals. At final, a number of 20 studies were included in our study, after the exclusion of duplicate papers.

## 5. Results

In the literature, we have not found any clinical trial highlighting the effects of *Spirulina* consumption in postmenopausal women suffering from metabolic syndrome. At the same time, no studies have been performed on the effects of *Spirulina* on metabolic syndrome, regardless of the category of included patients. On these conditions, we included in our study all the clinical trials that pointed out the effects of this cyanobacterium on each metabolic abnormality that is part of this syndrome. We found a number of 20 studies that included both women and men with metabolic abnormalities representing components of the metabolic syndrome. The main majority of the studies analyzed elderly subjects, including postmenopausal women. All the studies reported beneficial effects of *Spirulina*. However, further clinical trials are necessary to support the idea that *Spirulina* supplementation may aid postmenopausal women to manage the metabolic syndrome.

### 5.1. Spirulina Effects on Obesity

Recent reports suggested that *Spirulina* consumption may be useful in the management of obesity because of its effects on food absorption and appetite modulation [58]. Although the dietary supplementation with *Spirulina* seems to be a great opportunity to manage obesity, the scientific evidence in this field is still limited, and a small number of clinical trials were conducted on this topic. In addition, optimal posology is still to be established.

According to Fujimoto et al. [59], the primary mechanism of *Spirulina* in bodyweight reduction consists of the preventing of liver lipid accumulation, the decrease of oxidative stress and the reduction of the macrophages infiltration into visceral adipose tissue. Moreover, *Spirulina* is a rich source of essential amino acids, such as phenylalanine, which releases cholecystokinin. This molecule inhibits the appetite center at the level of the central nervous system and works as a bodyweight suppressant [60].

*Spirulina* also exhibits potent antioxidative effects. Antioxidants are useful in the management of obesity, and they act by various mechanisms: suppress food intake, inhibit the effects of lipase, modulate energy expenditure, inhibit adipocytes differentiation and regulate lipid metabolism [61].

Hernandez-Lepe et al. [62] reported that *Spirulina maxima* supplementation synergistically improved the effects of physical exercises on body composition in 25 obese and 27 overweight individuals. The body fat percentage was significantly reduced, and the maximal oxygen uptake was improved in subjects that associated physical exercises with *Spirulina* administration. These findings were more evident in obese than in overweight subjects.

Szulinska et al. [63] included 50 obese subjects in their clinical trial and administered them 2 g of *Spirulina* daily for three months. After this period, significant changes were observed in body composition: body mass index (BMI), waist circumference and total body mass decreased in the study group compared to controls, which received a placebo.

*Spirulina fusiformis* possesses a thin cellular wall, which makes this cyanobacterium an easy to digest nutraceutical food, with supplemental beneficial effects on body weight. Thirty individuals aged between 40 and 60 years were distributed into three groups: the first group received 2 g of *Spirulina fusiformis* daily for three months, the second group received 4 g of *Spirulina fusiformis* daily for three months, and the third group received placebo. The investigators observed that the reduction of the body weight was substantial in both study groups, in comparison with placebo [64].

*Spirulina maxima* have also been demonstrated to exert beneficial effects in body weight loss. After three months of daily administration, Miczke et al. [65] reported that regular consumption of this dietary supplement significantly improved BMI and total body weight. The dose of 500 mg of *Spirulina* administered twice/day for 12 weeks was also beneficial for overweight individuals. As Zeinalian et al. [66] reported, 1 g of *Spirulina* per day significantly reduced the appetite and decreased BMI and body weight in 64 obese subjects aged between 20 and 50 years. Furthermore, the lipid profile significantly improved.

A clinical trial conducted by Mazokopakis et al. [60] on Cretan subjects with non-alcoholic fatty liver disease reported that the supplementation with 6 g of *Spirulina platensis* for six months induced a significant reduction of the body weight, associated with multiple improvements of the metabolic abnormalities. A significant improvement in the health-related quality of life scale was also noticed.

In conclusion, *Spirulina* is a potent natural agent that can be used in the management of obesity. It induces both bodyweight loss and changes in body composition. Although there is not a wide range of studies conducted in this field, the age of the subjects included so far and the favorable results obtained may support the idea that *Spirulina* could be an effective treatment for obese or overweight women at menopause. Future studies are needed to optimize doses and the period of administration.

### 5.2. Spirulina Effects on Lipid Metabolism

Increasing evidence reported that doses between 1 and 10 g of *Spirulina* for at least 15 days might reduce the levels of atherogenic lipid molecules such as total cholesterol (TC), low-density lipoprotein cholesterol (LDL-C) or very-low-density lipoprotein cholesterol (VLDL-C). In addition, it increases the levels of high-density lipoprotein cholesterol (HDL-C), an antiatherogenic molecule [63,67].

H-b2 is a glycolipid contained by *Spirulina,* which inhibits pancreatic lipase activity in a dose-dependent manner [68]. Furthermore, phycocyanin has demonstrated its anti-lipase activity. On the other hand, phycocyanin and other bioactive molecules from *Spirulina* can increase the fecal excretion of cholesterol by decreasing its solubility [69].

Using HepG2 human cells, Ku et al. [70] have demonstrated that *Spirulina* is able to restore lipid metabolism balance by suppressing the expression of LDL receptor and downregulating the expression of 3-hydroxy-3-methyl-glutaryl-CoA reductase (HMGR). Furthermore, *Spirulina* inhibits lipid peroxidation and exhibits triglyceride-lowering properties due to the modulation of hepatic lipogenic gene expression and gut cholesterol absorption [70].

A clinical trial conducted by Torres-Duran et al. [71] has shown that the ingestion of 4.5 g of *Spirulina* daily for six weeks significantly improved the serum lipids levels. The treatment was administered to 36 individuals aged between 18 and 65 years, which did not modify their lifestyle of diet during the experimental period. After the treatment, LDL-C levels significantly decreased, while HDL-C and TC levels were dependent on triacylglycerol modifications.

Twelve individuals aged between 60 and 75 years were given 7.5 g of *Spirulina* daily for 24 weeks [72]. Biochemical assessment for plasma lipid levels has shown that TC, LDL-C and triglycerides levels recorded a significant decrease from only four weeks of the supplementation period. It is important to mention that these effects did not differ between individuals with normal cholesterol levels and hypercholesterolemic patients.

In contrast to the previous results that have shown significant improvements of serum LDL-C, TC and triglycerides levels after the consumption of *Spirulina,* Zeinalian et al. [66] reported that the intake of this cyanobacterium only partly modifies serum lipids. Sixty-four obese subjects, aged between 20 and 50 years, received 1 g of *Spirulina*/day during a period of 12 weeks. After this period, TC significantly reduced in the intervention group, while LDL-C and triglycerides did not suffer significant modifications compared to the control group.

A clinical trial conducted in 2016 on Korean elderly individuals has demonstrated that obesity may influence the hypolipidemic effects of *Spirulina*. In the non-obese group (BMI < 25 kg/m^2^), *Spirulina* supplementation (8 g daily, during a period of 12 weeks) induced a significant decrease in serum levels of TC and LDL-C, while in the obese group (BMI > 25 kg/m^2^) the results were not so obvious. Moreover, in the first group, the levels of interleukin 2 (IL-2) and antioxidant status levels recorded a significant increase after the treatment.

In conclusion, *Spirulina* also exerts beneficial effects on lipid metabolism. We observed that higher doses were administered for a longer period in order to obtain positive effects in comparison with the doses needed to control obesity.

### 5.3. Spirulina Effects on Serum Glucose and Insulin Resistance

Women with increased body mass index (BMI ≥ 24) or with later menopause are more predisposed to develop type 2 diabetes mellitus [73]. This disorder is characterized by metabolic and histological abnormalities due to a defect of insulin secretion or action, associated with oxidative stress [74]. Impaired glucose tolerance is characterized by insufficient insulin response in the peripheral target tissues, and it is a key point in the development of the metabolic syndrome.

Hu et al. [75] have isolated 11 peptides from *Spirulina platensis* using LC–MS/MS analysis and showed that three of them exhibited antidiabetic activities in vitro. This effect was possible due to their inhibition potential on dipeptidyl peptidase-4, α-amylase, and α-glucosidase.

*Spirulina* is rich in proteins and fibers, which induces decreased glucose absorption, associated with increased insulin secretion [76]. It has been demonstrated that *Spirulina* decreases serum levels of IL-6. This molecule can inhibit insulin-signaling molecules, such as insulin receptor substrate, leading to the suppression of glucose transporter type 4 (GLUT-4) translocation. Through this mechanism, the glucose uptake decreases in adipose depots and skeletal muscles [77].

The hypoglycemic effect of *Spirulina* has been demonstrated through several clinical trials. Mani et al. [78] observed the effects of *Spirulina* on non-insulin-dependent diabetes mellitus individuals. After two months of *Spirulina* supplementation, a significant reduction of glycated serum protein levels and blood sugar levels was observed. Furthermore, the lipid profile was improved, and a serious reduction of LDL-C, VLDL-C and triglycerides was remarked.

Twenty-five patients with type 2 diabetes mellitus were randomized to receive either 2 g/day of *Spirulina* or placebo. After two months, fasting blood glucose and HbA1c levels significantly decreased. Moreover, in patients with non-alcoholic fatty liver disease, the administration of *Spirulina* (6 g daily, for six months) significantly improved the homeostasis model assessment of insulin resistance (HOMA-IR) index [60].

Anitha et al. [79] conducted a study on non-insulin-dependent diabetic volunteers. After 12 weeks of *Spirulina* administration, a significant decrease in fasting blood glucose and glycosylated hemoglobin levels has been reported. Furthermore, in a group of seventeen HIV-infected subjects, which developed increased insulin-resistance, high doses of *Spirulina* (19 g daily for two months) significantly increased insulin sensitivity by 224.7%, in comparison with soybean supplementation [80]. In postmenopausal women, soybean is an alternative and natural treatment used for the management of menopause-related symptoms and pathologies. As we can observe, it also increases insulin sensitivity, but it is less efficient in comparison with *Spirulina.*

### 5.4. Spirulina Effects on Blood Pressure

*Spirulina* contains phycocyanin, a blue pigment with antioxidant activity, which is able to decrease the blood pressure values. Phycocyanin has been reported to enhance the expression of endothelial nitric oxide synthase in the aorta under the stimulation of adiponectin [81]. Five major peptides with hypotensive effects have been separated from *Spirulina* by high-performance liquid chromatography. Oral administration of Ile-Ala-Glu, Ala-Glu-Leu, Val-Ala-Phe, Ile-Ala-Glu, Ile-Ala-Pro-Gly (200 mg/kg) in hypertensive rats resulted in significant antihypertensive effects. The molecular mechanism of these peptides in hypertension includes the inhibition of angiotensin I-converting enzyme [82]. Furthermore, the peptides derived from *Spirulina* lower blood pressure values by inhibiting the RAA system [81].

Martinez-Samano et al. [83] performed a clinical trial, which included 16 patients with systemic arterial hypertension. Eight of them received *Spirulina maxima* and angiotensin-converting enzyme inhibitors, while the other eight received only angiotensin-converting enzyme inhibitors. After three months of daily administration, the researchers observed that systolic blood pressure significantly decreased in the group that received both medical treatment and *Spirulina.* Diastolic blood pressure did not register statistically significant modifications in the group that received both *Spirulina* and angiotensin-converting enzyme inhibitor.

According to the reports of Torres-Duran et al. [71], daily administration of 4.5 g of *Spirulina* for six weeks to overweight subjects may improve both systolic and diastolic values of the blood pressure. Moreover, the dietary supplementation with this product in patients with type 2 diabetes mellitus induced a significant reduction of the blood pressure. According to Lee et al. [84], the administration of 8 g of *Spirulina* daily for three months recorded excellent results in type 2 diabetes mellitus subjects.

Mickze et al. [65] also conducted a study to demonstrate the beneficial effects of *Spirulina* in overweight hypertensive patients. After three months of daily administration (2 g of *Spirulina*), the patients showed a significant reduction in systolic blood pressure.

These records demonstrate that regular consumption of *Spirulina* may improve blood pressure, especially in overweight patients. However, the posology and the optimal period of administration are still controversial. For this reason, further studies with longer duration and larger sample sizes are necessary to be conducted.

Table 1 synthesizes all the clinical studies regarding the effects of *Spirulina* on the components of metabolic syndrome, included in our systematic review.

## 6. Side Effects of *Spirulina* Administration

Very few reports regarding possible side effects of *Spirulina* consumption were made. However, in some cases, it may cause allergic reactions, headaches, sweating, or sleep disorders [91]. People with known allergies to various seafood or seaweeds should avoid the administration of *Spirulina* [92].

Lee et al. [93] reported that immunostimulatory dietary supplements, such as *Spirulina,* may exacerbate preexisting autoimmune disease. A patient that ingested *Spirulina platensis* presented a severe episode of dermatomyositis, while two patients that consumed Echinacea and *Spirulina platensis* developed pemphigus vulgaris only several days after the ingestion. The real mechanism of this phenomenon is still controversial, but the increased production of TNF-α may play a significant role.

Another side effect of *Spirulina* was reported by Mazokopakis et al. [94], which described the first case of acute rhabdomyolysis after the ingestion. A 17-year-old male developed anaphylaxis the first time he ingested a *Spirulina* tablet, according to Le et al. [95]. The diagnosis of *Spirulina* allergy can be simply established by using a skin prick test with dilutions of *Spirulina platensis* [95].

The consumption of this cyanobacterium can induce abdominal pain, nausea, or flatulence when it is consumed for the first time. If *Spirulina* is infected with various contaminants from the water, toxins, heavy metals or pollutants, it also gives rise to severe gastric perturbances. The safety of this product for pregnant or breastfeeding women is not well established, so it is recommended to be avoided during these periods.

## 7. Conclusions

*Spirulina* either grows in marine waters or can be cultured in special conditions and environments, and it has been proven that it possesses many therapeutic effects, including hypoglycemic, anti-obesity, antihyperlipidemic, and hypotensive activities. The administration of *Spirulina* has proved to be very efficient for the regulation of these metabolic abnormalities, which together constitute metabolic syndrome, one of the most prevalent pathologies in the elderly population, especially in the postmenopausal period. Despite there is a lack of studies focused on the effects of *Spirulina* in postmenopausal women with metabolic syndrome, we hope that our study paves the way for the researchers in this direction. In our opinion, given the beneficial effects of *Spirulina* on each component of the metabolic syndrome, the cumulative effect could fully target postmenopausal metabolic syndrome.

Although multiple clinical trials were conducted on the activity of *Spirulina* on metabolic abnormalities, the optimal dose and period of administration are still controversial and need a consensus. According to previous data, a minimal dose of 2 g of *Spirulina,* administered daily for a minimum of 2 months, recorded favorable results. High doses of *Spirulina* (18 or 19 g/day) administered for an extended period were also reported to have beneficial effects. According to the Food and Drug Administration, a recommended dosage for adults is usually in the range of 3–10 g day [96]. Multiple integrations with *Spirulina* and other nutraceuticals in menopausal women could influence the optimal dose, but there is no study conducted on this topic until now. *Spirulina* is generally recognized as safe for human consumption, but patients known with preexisting autoimmune diseases should avoid the ingestion of this cyanobacterium.

In conclusion, the seemingly conflicting results of the previous studies and the lack of studies focused on the effects of *Spirulina* on metabolic syndrome in both postmenopausal women and other categories of patients highlight the need for further comprehensive research in this field. In our opinion, based on the results obtained until now, the dietary supplementation of postmenopausal women with *Spirulina* could be a very attractive and efficient way to solve the metabolic abnormalities secondary to the sudden estrogen withdrawal.

## Figures and Tables

**Figure 1 marinedrugs-18-00651-f001:**
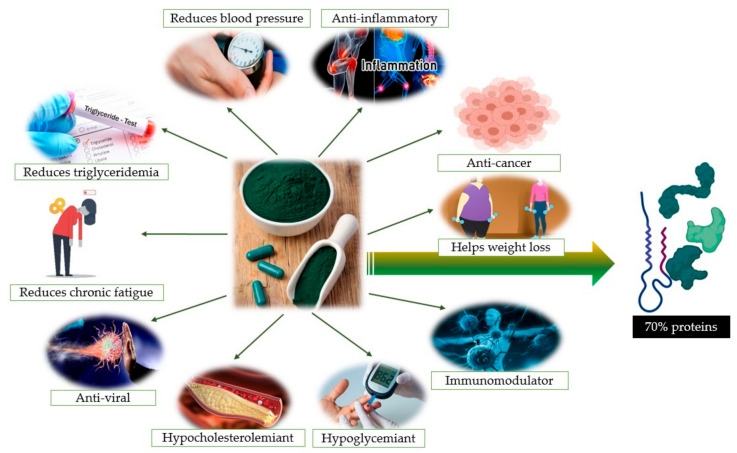
Health benefits of *Spirulina* supplementation [19].

**Figure 2 marinedrugs-18-00651-f002:**
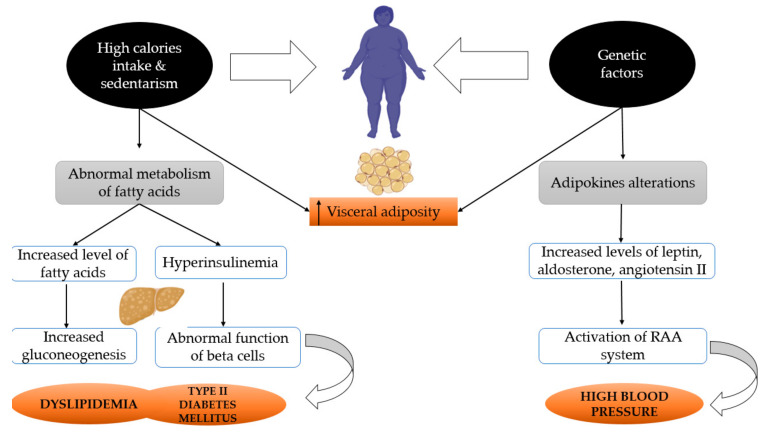
Mechanisms of metabolic abnormalities in postmenopausal women.

**Table 1 marinedrugs-18-00651-t001:** Studies regarding the effects of *Spirulina* on the components of metabolic syndrome.

Author, Year	Type of Study	*Spirulina* Effect	Participants	Posology	Results
Ramamoorthy et al. 1996 [64]	Double-blind placebo-controlled	- Bodyweight reduction - Hypocholesterolemiant effect	30 overweight patients aged between 40 and 60 years, with high blood cholesterol levels	Group A—2 g of *Spirulina* daily—3 months Group B—4 g of *Spirulina* daily—3 months Group C—placebo	- Mean initial weights significantly decreased in group A and B compared to group C - Total cholesterol levels significantly decreased in group B - Serum triglyceride, LDL, VLDL levels significantly decreased in group A and B in comparison with group C - HDL significantly increased in group A and B in comparison with group C
Kim et al. 2003 [72]	Randomized clinical trial	- Improves lipid metabolism - Improves antioxidant capacity and immune function	6 males and 6 females aged between 60 and 75 years	7.5 g of *Spirulina* daily—6 months	- Serum levels of triglycerides, total cholesterol and LDL-cholesterol significantly decreased - Immune function and antioxidant capacity significantly improved - No modification was recorded for anthropometric parameters
Samuels et al. 2004 [85]	Double-blind placebo-controlled	- Hypocholesterolemiant effect	23 patients aged between 2 and 13 years with nephrotic syndrome and secondary hyperlipidemia	Group 1—medication—2 months Group 2—medication and 1 g of *Spirulina* daily—2 months	- Anthropometric parameters were normal compared with healthy individuals - Triglycerides, total cholesterol and LDL-cholesterol levels significantly decreased in group 2 - LDL-C:HDL-C ratio decreased more significant in group 2
Parikh et al. 2004 [76]	Double-blind placebo-controlled	- Hypoglycemia effects - Hypocholesterolemiant effects	25 patients with type 2 diabetes mellitus	Group 1—2 g of *Spirulina* daily—2 months Group 2—placebo	- Fasting blood glucose and postprandial blood glucose levels significantly decreased - HbA1c levels showed a significant reduction - Total cholesterol, LDL-cholesterol decreased, and HDL-cholesterol increased in group 1 - apo B decreased, and apo A1 increased
Torres-Duran et al. 2007 [71]	Prospective study	- Antihyperlipidemic effects - Antihypertensive effects	16 men and 20 women aged between 18 and 65 years	All the participants received 4.5 g of *Spirulina* daily for 6 weeks	- Significant differences were registered in triacylglycerols, total cholesterol and HDL-cholesterol levels - Lower values of systolic and diastolic blood pressured were observed after the treatment
Lee et al. 2008 [84]	Double-blind placebo-controlled	- Bodyweight reduction - Antihyperlipidemic effects - Antihypertensive effects	15 males and 16 females aged between 30 and 70 years, with type 2 diabetes mellitus	Study group—8 g of *Spirulina* daily—3 months Control group—placebo	- Anthropometric parameters were not significantly changed for both *Spirulina* and control groups. - Plasma triglycerides and diastolic blood pressure significantly decreased in the study group in comparison with controls, - Plasma adiponectin increased - The individuals with higher initial triglyceride serum level showed a more significant reduction in weight, abdominal fat thickness, waist-to-hip ratio, triglyceride levels, atherogenic index, and diastolic blood pressure
Kaur et al. 2008 [86]	Double-blind placebo-controlled	- Hypoglycemic effects - Antihyperlipidemic effects	60 males aged between 40 and 60 years, with type 2 diabetes mellitus	Group E1—1 g of *Spirulina* daily—2 months Group E2—2 g of *Spirulina* daily—2 monthsGroup C—placebo	- The mean fasting and postprandial blood glucose level significantly decreased in group E1 and E2 - Total cholesterol, triglycerides, LDL-cholesterol and VLDL-cholesterol levels significantly decreased in E groups.
Park et al. 2008 [77]	Randomized double-blind, placebo-controlled study	- Antihyperlipidemic effects - Improves immune status - Antioxidant effects	78 subjects aged between 60 and 87 years	Study group—8 g of *Spirulina* daily—16 weeks Control group—placebo	- Total cholesterol significantly decreased in both males and women - IL-6 serum levels decreased - In female participants was observed a significant increase in IL-2 level and SOD activity
Anitha et al. 2010 [79]	Randomized double-blind, placebo-controlled study	- Hypoglycemic effects - Antihyperlipidemic effects	160 male volunteers, non-insulin-dependent diabetics	Group 1—placebo Group 2—dietary regimen + 1 g *Spirulina—*12 weeks Group 3—diet and drugs + 1 g *Spirulina—*12 weeks Group 4—diet, drugs, and insulin + 1 g *Spirulina—*12 weeks	- The levels of fasting blood glucose and glycosylated hemoglobin significantly decreased - HDL-cholesterol increased - Lipid profile significantly improved
Azabji-Kenfack et al. 2011 [80]	Randomized double-blind, placebo-controlled study	- Improves insulin sensitivity	33 insulin-resistant HIV-infected patients	17 subjects received 19 g of *Spirulina* daily for 2 months 16 subjects received soybean	- 100% vs. 69% of subjects on *Spirulina* versus soybean, respectively, improved their insulin sensitivity
Mazokopakis et al. 2014 [60]	Prospective study	- Hypocholesterolemiant effects - Bodyweight loss	13 men and 2 women aged between 29 and 62 years, with non-alcoholic fatty liver disease	6 g of *Spirulina* daily for 6 months	- Triglycerides, LDL-cholesterol, total cholesterol and TC:HDL-cholesterol ration significantly decreased - Significant reduction in weight and HOMA-IR index were reported
Ismail et al. 2014 [87]	Randomized double-blind, placebo-controlled study	- Improves lipid metabolism - Antioxidant effects	30 subjects with chronic obstructive pulmonary disease and 20 healthy controls	Group 1—1 g of *Spirulina* daily for 60 days Group 2—2 g of *Spirulina* daily for 60 days Group 3—placebo	- Serum contents of total cholesterol, lipid hydroperoxide, and malondialdehyde were significantly decreased - No differences were observed in serum triglyceride levels - SOD, GSH and GST activity significantly increased
Ngo-Matip et al. 2014 [88]	Prospective, single-blind, randomized study	- Improves lipid metabolism	169 HIV-antiretroviral naïve subjects, with metabolic alterations	Group 1—Local diet + *Spirulina* supplementation—12 months Group 2—Local diet only	- In group 1 was observed a significant increase in HDL-cholesterol and a significant decrease in total cholesterol, LDL-cholesterol and triglyceride levels
Mani et al. 2015	Prospective study	- Hypoglycemiant effects - Antihyperlipidemic effects	15 non-insulin-dependent diabetes mellitus subjects	2 g of *Spirulina—*2 months	- Blood sugar levels and glycated serum protein levels significantly decreased - Total cholesterol, free fatty acids, triglycerides, LDL-cholesterol, VLDL-cholesterol, HDL-C/LDL-C significantly degreased
Park et al. 2016 [89]	Randomized double-blind, placebo-controlled study	- Antihyperlipidemic effects - Antioxidant effects - Enhances the immune system	78 patients aged 60–87 years	Group 1—8 g of *Spirulina* daily—12 weeks Group 2—placebo	- Serum levels of LDL-cholesterol and triglycerides significantly decreased - IL-2 increased - Total antioxidant status level significantly increased
Micze et al. 2016 [65]	Randomized double-blind, placebo-controlled study	- Bodyweight loss - Hypotensive effects	40 hypertensive individuals, lacking evidence of cardiovascular disease	Group 1—2 g of *Spirulina* daily—3 months Group 2—placebo	- The subjects from group 1 showed significant reductions in BMI and weight - Systolic blood pressure and stiffness index significantly decreased in the study group
Zeinalian et al. 2017 [66]	Randomized double-blind, placebo-controlled study	- Improves lipid metabolism - Decreases appetite and body weight	64 obese subjects aged between 20 and 50 years	29 subjects received 1 g of *Spirulina* daily—12 weeks 27 subjects—control group—received placebo	- Bodyweight and body mass index significantly decreased in *Spirulina*-treated group; appetite decreased in the study group - Serum VEGF, LDL-cholesterol and triglycerides levels did not change significantly after intervention - HDL-cholesterol significantly increased in both groups
Hernandez-Lepe et al. 2018 [62]	Randomized double-blind, crossover controlled trial	- Improves the effects of physical exercises on body composition and cardiorespiratory fitness parameters in obese and overweight patients	27 overweight and 25 obese males	Group 1—physical exercise program twice weekly + 4.5 g of *Spirulina*—6 weeks Group 2—physical exercise program twice weekly Group 3—no physical exercise program + 4.5 g of *Spirulina*—6 weeks Group 4—control group	- *Spirulina* improved the maximal oxygen uptake and decreased the body fat percentage - The time to reach fatigue and the onset of blood lactate accumulation were improved in group 1 and group 3
Martinez-Samano et al. 2018 [83]	Prospective, randomized, parallel pilot study	- Antihypertensive effects - Antioxidant effects	16 patients with systemic arterial hypertension undergoing treatment with angiotensin-converting enzyme	Group 1—angiotensin-converting enzyme + 4.5 g of *Spirulina* for 12 weeks Group 2—placebo + angiotensin-converting enzyme	- Systolic blood pressure significantly decreased - sVCAM-1, E-selectin and endothelin-1 levels significantly decreased - Glutathione peroxidase activity and oxidized glutathione levels decreased
Pancholi et al. 2019 [90]	Prospective study	- Helps body weight loss	100 obese patients aged between 18 and 70 years (50 women and 50 males)	Group 1—non-*Spirulina* supplementation Group 2—5 g of *Spirulina* daily—45 days	- The subjects from group 2 lost 3 kg weight and improved sleeping quality

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
