# Peer review of "Are There Any Beneficial Effects of Spirulina Supplementation for Metabolic Syndrome Components in Postmenopausal Women?"

_marinedrugs, 2020, doi:10.3390/md18120651_

Round 1
Reviewer 1 Report
Review:
Manuscript ID: marinedrugs-1023767
Type of manuscript: Review
Title: Are There Any Beneficial Effects of Spirulina Supplementation for
Metabolic Syndrome Components in Postmenopausal Women?
The authors reviewed the health aspects of Spirulina with special focus to postmenopausal women.
The authors analyzed 20 publications and described the effects of Spirulina extracts to human health.
The manuscript is well written and the selection of publications is good. They collected quite some information. However, I don’t see the connection to the postmenopausal women, since the studies described, focused on special groups with singular health issues, like obesity or diabetes. To draw conclusions to a complex symptom seems farfetched. The authors have to describe in detail the impacts to the focus group.
Furthermore, the manuscript needs extensive rewriting.
1.: Spirulina is not a microalga, it is a cyanobacterium. Please change that throughout the manuscript.
2.: Spirulina can not fix nitrogen. Please change.
3.: Delete the two rat experiments from the manuscript.
4.: Please read the manuscript carefully and delete repetitive passages and shorten the manuscript.
See the marked passages in the attached pdf file. Especially introduction to each paragraph (or at least the majority of them) can be deleted. I would recommend ending each paragraph with a summary, to emphasize the findings regarding the topic of the manuscript.
6.: Rewrite Paragraph 193-201: Shift sentences to make it more consistent. Phycocyanobilin is part of the phycobilins, see structure of pigments in cyanobacteria.
- Citation 21 seems a bit arbitrary. Please find other reviews regarding this topic.
- line 147 more up to previous paragraph
- line 264: unclear no effect between placebo and spirulina treated patients or no differences between both spirulina treated patients?
- I don’t think that spirulina was frequently used since ancient times. See line 411

Author Response
Dear Reviewer,
We thank you for your cooperation and we appreciate you taking the time to analyze our work. Following your comments and suggestions, we made some revisions to our paper, in order to have a clearer presentation of our results, as follows:
Point 1: The manuscript is well written and the selection of publications is good. They collected quite some information. However, I don’t see the connection to the postmenopausal women, since the studies described, focused on special groups with singular health issues, like obesity or diabetes. To draw conclusions to a complex symptom seems farfetched. The authors have to describe in detail the impacts to the focus group.
Response 1: The main objective of this article was to gather the existing evidence regarding the effects of Spirulina on each metabolic abnormality that is part of the metabolic syndrome. There are no studies in the literature on the effects of Spirulina on menopause-related metabolic abnormalities, which is why, in this article, we wanted to emphasize the possibility that this cyanobacterium become a therapeutic tool for this category of patients, with increasing incidences of metabolic syndrome.
Point 2: Spirulina is not a microalga, it is a cyanobacterium. Please change that throughout the manuscript.
Response 2: We made the required modifications
Point 3: Spirulina can not fix nitrogen. Please change
Response 3: Spirulina can fix nitrogen, and if the environment is poor in nitrogen, Spirulina changes its characteristics. We added another reference source that explains the effects of nitrogen on Spirulina.
Point 4: Delete the two rat experiments from the manuscript
Response 4: We deleted the two rat experiments
Point 5: Please read the manuscript carefully and delete repetitive passages and shorten the manuscript. See the marked passages in the attached pdf file. Especially introduction to each paragraph (or at least the majority of them) can be deleted. I would recommend ending each paragraph with a summary, to emphasize the findings regarding the topic of the manuscript
Response 5: We corrected the highlighted passages and deleted the repetitive ones. Moreover, we excluded the introduction to many paragraphs and shortened our paper. Also, we tried to end the paragraphs regarding the effects of Spirulina on each metabolic abnormality from the metabolic syndrome with a conclusion, and to create a link with postmenopausal period.
Point 6: Rewrite Paragraph 193-201: Shift sentences to make it more consistent. Phycocyanobilin is part of the phycobilins, see structure of pigments in cyanobacteria.
Response 6: We made the requested modifications
Point 7: line 147 more up to previous paragraph; line 264: unclear no effect between placebo and spirulina treated patients or no differences between both spirulina treated patients?
Response 7: We made the requested modifications and we explained better the results of this study.
Besides, we made some changes of the English language in order to make the message easier to understand.
Reviewer 2 Report
Please see the attached file below.

Author Response
Dear Sir or Madam,
We thank you for your cooperation and we appreciate you taking the time to analyze our work. Following your comments and suggestions, we made some revisions to our paper, in order to have a clearer presentation of our results, as follows:
Point 1: There is some evidence that is possible a multiple integration with Spirulina and other nutraceuticals in menopausal women. This aspect could impact the optimal dose. Please discuss this intriguing aspect in your review paper.
Response 1: We found only one study discussing the comparative effects of Spirulina and soybean on the glucidic metabolism, but no studies regarding a multiple integration with Spirulina and other nutraceuticals in menopausal women were conducted yet. We mentioned this aspect in our paper, which is also very interesting to study in the future.
Point 2: Please focus only on human studies.
Response 2: We excluded the animal experiment from our review
Point 3: Please shorten the manuscript particularly the introduction of each paragraph, then focusing on Spirulina effects in patients.
Response 3: We deleted the repetitive passage. Moreover, we excluded the introduction to many paragraphs and shortened our paper. Also, we tried to end the paragraphs regarding the effects of Spirulina on each metabolic abnormality from the metabolic syndrome with a conclusion, and to create a link with postmenopausal period.
Besides, we made some changes of the English language in order to make the message easier to understand.